# Editable Neural Networks

**Anton Sinitsin**[*]
Yandex
National Research University
Higher School of Economics
`ant.sinitsin@gmail.com`

**Vsevolod Plokhotnyuk**[*]
National Research University
Higher School of Economics
`vsevolod-pl@yandex.ru`

**Dmitry Pyrkin**[*]
National Research University
Higher School of Economics
`alagaster@yandex.ru`

**Sergei Popov**
Yandex
`sapopov@yandex-team.ru`

**Artem Babenko**
Yandex
National Research University
Higher School of Economics
`artem.babenko@phystech.edu`

## Abstract

These days deep neural networks are ubiquitously used in a wide range of tasks, from image classification and machine translation to face identification and self-driving cars. In many applications, a single model error can lead to devastating financial, reputational and even life-threatening consequences. Therefore, it is crucially important to correct model mistakes quickly as they appear. In this work, we investigate the problem of neural network editing — how one can efficiently patch a mistake of the model on a particular sample, without influencing the model behavior on other samples. Namely, we propose Editable Training, a model-agnostic training technique that encourages fast editing of the trained model. We empirically demonstrate the effectiveness of this method on large-scale image classification and machine translation tasks.

## 1 Introduction

Deep neural networks match and often surpass human performance on a wide range of tasks including visual recognition (Krizhevsky et al. (2012); D. C. Ciresan (2011)), machine translation (Hassan et al. (2018)) and others (Silver et al. (2016)). However, just like humans, artificial neural networks sometimes make mistakes. As we trust them with more and more important decisions, the cost of such mistakes grows ever higher. A single misclassified image can be negligible in academic research but can be fatal for a pedestrian in front of a self-driving vehicle. A poor automatic translation for a single sentence can get a person arrested (Hern (2018)) or ruin company's reputation.

Since mistakes are inevitable, deep learning practitioners should be able to adjust model behavior by correcting errors as they appear. However, this is often difficult due to the nature of deep neural networks. In most network architectures, a prediction for a single input depends on all model parameters. Therefore, updating a neural network to change its predictions on a single input can decrease performance across other inputs.

Currently, there are two workarounds commonly used by practitioners. First, one can re-train the model on the original dataset augmented with samples that account for the mistake. However, this is computationally expensive as it requires to perform the training from scratch. Another solution is to use a manual cache (e.g. lookup table) that overrules model predictions on problematic samples.

---

[*]Equal contribution

While being simple, this approach is not robust to minor changes in the input. For instance, it will not generalize to a different viewpoint of the same object or paraphrasing in natural language processing tasks.

In this work, we present an alternative approach that we call Editable Training. This approach involves training neural networks in such a way that the trained parameters can be easily edited afterwards. Editable Training employs modern meta-learning techniques (Finn et al. (2017)) to ensure that model's mistakes can be corrected without harming its overall performance. With thorough experimental evaluation, we demonstrate that our method works on both small academical datasets and industry-scale machine learning tasks. We summarize the contributions of this study as follows:

- We address a new problem of fast editing of neural network models. We argue that this problem is extremely important in practice but, to the best of our knowledge, receives little attention from the academic community.

- We propose Editable Training — a model-agnostic method of neural network training that learns models, whose errors can then be efficiently corrected.[1]

- We extensively evaluate Editable Training on large-scale image classification and machine translation tasks, confirming its advantage over existing baselines.

## 2   RELATED WORK

In this section, we aim to position our approach with respect to existing literature. Namely, we explain the connections of Editable Neural Networks with ideas from prior works.

**Meta-learning** is a family of methods that aim to produce learning algorithms, appropriate for a particular machine learning setup. These methods were shown to be extremely successful in a large number of problems, such as few-shot learning (Finn et al. (2017); Nichol et al. (2018)), learnable optimization (Andrychowicz et al. (2016)) and reinforcement learning (Houthooft et al. (2018)). Indeed, Editable Neural Networks also belong to the meta-learning paradigm, as they basically "learn to allow effective patching". While neural network correction has significant practical importance, we are not aware of published meta-learning works, addressing this problem.

**Catastrophic forgetting** is a well-known phenomenon arising in the problem of lifelong/continual learning (Ratcliff (1990)). For a sequence of learning tasks, it turns out that after deep neural networks learn on newer tasks, their performance on older tasks deteriorates. Several lines of research address overcoming catastrophic forgetting. The methods based on Elastic Weight Consolidation (Kirkpatrick et al. (2016)) update model parameters based on their importance to the previous learning tasks. The rehearsal-based methods (Robins (1995)) occasionally repeat learning on samples from earlier tasks to "remind" the model about old data. Finally, a line of work (Garnelo et al. (2018); Lin et al. (2019)) develops specific neural network architectures that reduce the effect of catastrophic forgetting. The problem of efficient neural network patching differs from continual learning, as our setup is not sequential in nature. However, correction of model for mislabeled samples must not affect its behavior on other samples, which is close to overcoming catastrophic forgetting task.

**Adversarial training.** The proposed Editable Training also bears some resemblance to the adversarial training (Goodfellow et al. (2015)), which is the dominant approach of adversarial attack defense. The important difference here is that Editable Training aims to learn models, whose behavior on some samples can be efficiently corrected. Meanwhile, adversarial training produces models, which are robust to certain input perturbations. However, in practice one can use Editable Training to efficiently cover model vulnerabilities against both synthetic (Szegedy et al. (2013); Yuan et al. (2017); Ebrahimi et al. (2017); Wallace et al. (2019)) and natural (Hendrycks et al. (2019)) adversarial examples.

---

[1]The source code is available online at `https://github.com/xtinkt/editable`

## 3 EDITING NEURAL NETWORKS

In order to measure and optimize the model's ability for editing, we first formally define the operation of editing a neural network. Let $f(x, \theta)$ be a neural network, with $x$ denoting its input and $\theta$ being a set of network parameters. The parameters $\theta$ are learned by minimizing a task-specific objective function $\mathcal{L}_{base}(\theta)$, e.g. cross-entropy for multi-class classification problems.

Then, if we discover mistakes in the model's behavior, we can patch the model by changing its parameters $\theta$. Here we aim to change model's predictions on a subset of inputs, corresponding to misclassified objects, without affecting other inputs. We formalize this goal using the *editor function*: $\hat{\theta} = Edit(\theta, l_e)$. Informally, this is a function that adjusts $\theta$ to satisfy a given constraint $l_e(\hat{\theta}) \leq 0$, whose role is to enforce desired changes in the model's behavior.

For instance, in the case of multi-class classification, $l_e$ can guarantee that the model assigns input $x$ to the desired label $y_{ref}$: $l_e(\hat{\theta}) = \max_{y_i} \log p(y_i|x, \hat{\theta}) - \log p(y_{ref}|x, \hat{\theta})$. Under such definition of $l_e$, the constraint $l_e(\hat{\theta}) \leq 0$ is satisfied iff $\arg\max_{y_i} \log p(y_i|x, \hat{\theta}) = y_{ref}$.

To be practically feasible, the editor function must meet three natural requirements:

- **Reliability:** the editor must guarantee $l_e(\hat{\theta}) \leq 0$ for the chosen family of $l_e(\cdot)$;

- **Locality:** the editor should minimize influence on $f(\cdot, \hat{\theta})$ outside of satisfying $l_e(\hat{\theta}) \leq 0$;

- **Efficiency:** the editor should be efficient in terms of runtime and memory;

Intuitively, the editor locality aims to minimize changes in model's predictions for inputs unrelated to $l_e$. For classification problem, this requirement can be formalized as minimizing the difference between model's predictions over the "control" set $X_c$: $\underset{x \in X_c}{E} \#[f(x, \hat{\theta}) \neq f(x, \theta)] \to \min$.

### 3.1 GRADIENT DESCENT EDITOR

A natural way to implement $Edit(\theta, l_e)$ for deep neural networks is using gradient descent. Parameters $\theta$ are shifted against the gradient direction $-\alpha\nabla_\theta l_e(\theta)$ for several iterations until the constraint $l_e(\theta) \leq 0$ is satisfied. We formulate the SGD editor with up to $k$ steps and learning rate $\alpha$ as:

$$Edit_\alpha^k(\theta, l_e, k) = \begin{cases} \theta, & \text{if } l_e(\theta) \leq 0 \text{ or } k = 0 \\ Edit_\alpha^{k-1}(\theta - \alpha \cdot \nabla_\theta l_e(\theta), l_e), & \text{otherwise} \end{cases} \quad (1)$$

The standard gradient descent editor can be further augmented with momentum, adaptive learning rates (Duchi et al. (2010); Zeiler (2012)) and other popular deep learning tricks (Kingma & Ba (2014); Smith & Topin (2017)). One technique that we found practically useful is Resilient Backpropagation: RProp, SignSGD by Bernstein et al. (2018) or RMSProp by Tieleman & Hinton (2012). We observed that these methods produce more robust weight updates that improve locality.

### 3.2 EDITABLE TRAINING

The core idea behind Editable Training is to enforce the model parameters $\theta$ to be "prepared" for the editor function. More formally, we want to learn such parameters $\theta$, that the editor $Edit(\theta, l_e)$ is reliable, local and efficient, as defined in above.

Our training procedure employs the fact that Gradient Descent Editor (1) is differentiable w.r.t. $\theta$. This well-known observation (Finn et al. (2017)) allows us to optimize through the editor function directly via backpropagation (see Figure 1).

Editable Training is performed on minibatches of constraints $l_e \sim p(l_e)$ (e.g. images and target labels). First, we compute the edited parameters $\hat{\theta} = Edit(\theta, l_e)$ by applying up to $k$ steps of gradient descent (1). Second, we compute the objective that measures locality and efficiency of the editor function:

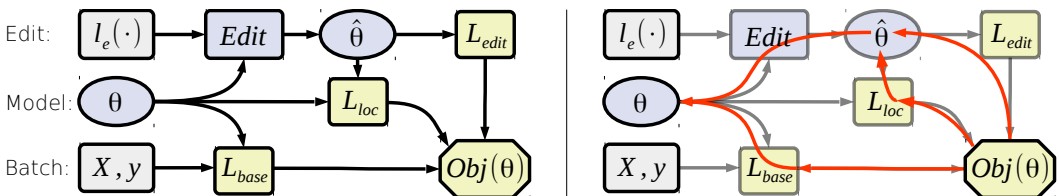

Figure 1: A high-level scheme of editable training: (left) forward pass, (right) backward pass.

$$Obj(\theta, l_e) = \mathcal{L}_{base}(\theta) + c_{edit} \cdot \mathcal{L}_{edit}(\theta) + c_{loc} \cdot \mathcal{L}_{loc}(\theta) \tag{2}$$

$$\mathcal{L}_{edit}(\theta) = max(0, l_e(Edit_\alpha^k(\theta, l_e)) \tag{3}$$

$$\mathcal{L}_{loc}(\theta) = \mathop{E}_{x \sim p(x)} D_{KL}(p(y|x, \theta)||p(y|x, Edit_\alpha^k(\theta, l_e))) \tag{4}$$

Intuitively, $\mathcal{L}_{edit}(\theta)$ encourages reliability and efficiency of the editing procedure by making sure the constraint is satisfied in under $k$ gradient steps. The final term $\mathcal{L}_{loc}(\theta)$ is responsible for locality by minimizing the KL divergence between the predictions of original and edited models.

We use hyperparameters $c_{edit}, c_{loc}$ to balance between the original task-specific objective, editor efficiency and locality. Setting both of them to large positive values would cause the model to sacrifice some of its performance for a better edit. On the other hand, sufficiently small $c_{edit}, c_{loc}$ will not cause any deterioration of the main training objective while still improving the editor function in all our experiments (see Section 4). We attribute this to the fact that neural networks are typically overparameterized. Most neural networks can accommodate the edit-related properties and still have enough capacity to optimize $Obj(\theta, l_e)$. The learning step $\alpha$ and other optimizer parameters (e.g. $\beta$ for RMSProp) are trainable parameters of Editable Training and we optimize them explicitly via gradient descent.

## 4 EXPERIMENTS

In this section, we extensively evaluate Editable Training on several deep learning problems and compare it to existing alternatives for efficient model patching.

### 4.1 TOY EXPERIMENT: CIFAR-10

First, we experiment on image classification with the small CIFAR-10 dataset with standard train/test splits (Krizhevsky et al.). The training dataset is further augmented with random crops and random horizontal flips. All models trained on this dataset follow the ResNet-18 (He et al. (2015)) architecture and use the Adam optimizer (Kingma & Ba (2014)) with default hyperparameters.

Our baseline is ResNet-18 (He et al. (2015)) neural network trained to minimize the standard cross-entropy loss without Editable Training. This model provides **6.3%** test error rate at convergence.

**Comparing editor functions.** As a preliminary experiment, we compare several variations of editor functions for the baseline model without Editable Training. We evaluate each editor by applying $N=1000$ edits $l_e$. Each edit consists of an image from the test set assigned with a random (likely incorrect) label uniformly chosen from 0 to 9. After $N$ independent edits, we compute three following metrics over the entire test set:

- **Drawdown** — mean absolute difference of classification error before and after performing an edit. Smaller drawdown indicates better editor locality.
- **Success Rate** — a rate of edits, for which editor succeeds in under $k=10$ gradient steps.
- **Num Steps** — an average number of gradient steps needed to perform a single edit.

- **Gradient Descent (GD)** — standard gradient descent.

| Editor Function | GD | Scaled GD | RProp | RMSProp | Momentum | Adam |
|---|---|---|---|---|---|---|
| Drawdown | 3.8% | 2.81% | 1.99% | **1.77%** | 2.42% | 19.4% |
| Success Rate | 98.8% | 99.1% | 100% | 100% | 96.0% | 100% |
| Num Steps | 3.54 | 3.91 | 2.99 | 3.11 | 5.60 | 3.86 |

Table 1: Comparison of different editor functions on the CIFAR10 dataset with the baseline ResNet18 model trained without Editable Training.

- **Scaled GD** — like GD, but the learning rate is divided by the global gradient norm from the first gradient step.
- **RProp** — like GD, but the algorithm only uses the sign of gradients: $\theta - \alpha \cdot sign(\nabla_\theta l_e(\theta))$.
- **RMSProp** — like GD, but the learning rate for each individual parameter is divided by $\sqrt{rms_t + \epsilon}$ where $rms_0 = [\nabla_\theta l_e(\theta_0)]^2$ and $rms_{t+1} = \beta \cdot rms_t + (1 - \beta) \cdot [\nabla_\theta l_e(\theta)]^2$.
- **Momentum GD** — like GD, but the update follows the accumulated gradient direction $\nu$: $\nu_0 = 0; \nu_{t+1} = \alpha \cdot \nabla_\theta l_e(\theta_0) + \mu \cdot \nu_t$.
- **Adam** — adaptive momentum algorithm as described in Kingma & Ba (2014) with tunable $\alpha, \beta_1, \beta_2$. To prevent Adam from replicating RMSProp, we restrict $\beta_1$ to $[0.1, 1.0]$ range.

For each optimizer, we tune all hyperparameters (e.g. learning rate) to optimize locality while ensuring that editor succeeds in under $k = 10$ steps for at least $95\%$ of edits. We also tune the editor function by limiting the subset of parameters it is allowed to edit. The ResNet-18 model consists of six parts: initial convolutional layer, followed by four "chains" of residual blocks and a final linear layer that predicts class logits. We experimented with editing the whole model as well as editing each individual "chain", leaving parameters from other layers fixed. For each editor Table 1 reports the numbers, obtained for the subset of editable parameters, corresponding to the smallest drawdown. For completeness, we also report the drawdown of Gradient Descent and RMSProp for different subsets of editable parameters in Table 2.

| Editable Layers | Whole Model | Chain 1 | Chain 2 | Chain 3 | Chain 4 |
|---|---|---|---|---|---|
| Gradient Descent | **3.8%** | 18.3% | 7.7% | 5.3% | 4.76% |
| RMSProp | 2.29% | 22.8% | 1.85% | **1.77%** | 1.99% |

Table 2: Mean Test Error Drawdown when editing different ResNet18 layers on CIFAR10.

Table 1 and Table 2 demonstrate that the editor function locality is heavily affected by the choice of editing function even for models trained without Editable Training. Both RProp and RMSProp significantly outperform the standard Gradient Descent while Momentum and Adam show smaller gains. In fact, without the constraint $\beta_1 > 0.1$ the tuning procedure returns $\beta_1 = 0$, which makes Adam equivalent to RMSProp. We attribute the poor performance of Adam and Momentum to the fact that most methods only make a few gradient steps till convergence and the momentum term cannot accumulate the necessary statistics.

**Editable Training.** Finally, we report results obtained with Editable Training. On each training batch, we use a single constraint $l_e(\hat{\theta}) = \max_{y_i} \log p(y_i|x, \hat{\theta}) - \log p(y_{ref}|x, \hat{\theta})$, where $x$ is sampled from the train set and $y_{ref}$ is a random class label (from 0 to 9). The model is then trained by directly minimizing objective (2) with $k$=10 editor steps and all other parameters optimized by backpropagation.

We compare our Editable Training against three baselines, which also allow efficient model correction. The first natural baseline is Elastic Weight Consolidation (Kirkpatrick et al. (2016)): a technique that penalizes the edited model with the squared difference in parameter space, weighted by the importance of each parameter. Our second baseline is a semi-parametric Deep k-Nearest Neighbors (**DkNN**) model (Papernot & McDaniel (2018)) that makes predictions by using $k$ nearest neighbors in the space of embeddings, produced by different CNN layers. For this approach, we edit the model by flipping labels of nearest neighbors until the model predicts the correct class.

Finally we compare to alternative editor function inspired by Conditional Neural Processes (CNP) (Garnelo et al. (2018)) that we refer to as **Editable+CNP**. For this baseline, we train a specialized

CNP model architecture that performs edits by adding a special *condition vector* to intermediate activations. This vector is generated by an additional "encoder" layer. We train the CNP model to solve the original classification problem when the condition vector is zero (hence, the model behaves as standard ResNet18) and minimize $\mathcal{L}_{edit}$ and $\mathcal{L}_{loc}$ when the condition vector is applied.

After tuning the CNP architecture, we obtained the best performance when the condition vector is computed with a single ResNet block that receives the image representation via activations from the third residual chain of the main ResNet-18 model. This "encoder" also conditions on the target class $y_{ref}$ with an embedding layer (lookup table) that is added to the third chain activations. The resulting procedure becomes the following: first, apply encoder to the edited sample and compute the condition vector, then add this vector to the third layer chain activations for all subsequent inputs.

| Training Procedure | Editor Function | Editable Layers | Test Error Rate | Test Error Drawdown | Success Rate | Num Steps |
|---|---|---|---|---|---|---|
| Baseline Training | GD | All | 6.3% | 3.8% | 98.8% | 3.54 |
| | RMSProp | Chain 3 | 6.3% | 1.77% | 100% | 3.11 |
| Editable $c_{loc} = 0.01$ | GD | All | 6.34% | 1.42% | 100% | 3.39 |
| | GD | Chain 3 | 6.28% | 1.44% | 100% | 2.82 |
| | RMSProp | Chain 3 | 6.31% | **0.86%** | 100% | 4.13 |
| Editable $c_{loc} = 0.1$ | RMSProp | Chain 3 | 7.19% | **0.65%** | 100% | 4.76 |
| Editable+CNP (best) | Cond. vector | Chain 3 | 6.33% | 1.06% | 98.9% | n/a |
| Baseline Training | GD+EWC | Chain 3 | 6.3% | 1.92% | 100% | 3.88 |
| Baseline Training | RMSProp+EWC | Chain 3 | 6.3% | 1.24% | 98.1% | 4.03 |
| DkNN $k = 10$ | Flip Labels | n/a | 6.36% | 1.76% | 100% | n/a |
| DkNN $k = 100$ | Flip Labels | n/a | 7.04% | 1.05% | 100% | n/a |

Table 3: Editable Training of ResNet18 on CIFAR10 dataset with different editor functions.

Table 3 demonstrates two advantages of Editable Training. First, with $c_{loc}$=0.01 it is able to reduce drawdown (compared to models trained without Editable Training) while having no significant effect on test error rate. Second, editing Chain 3 alone is almost as effective as editing the whole model. This is important because it allows us to reduce training time, making Editable Training $\approx 2.5$ times slower than baseline training. Note, Editable+CNP turned out to be almost as effective as models trained with gradient-based editors while being simpler to implement.

## 4.2 ANALYZING EDITED MODELS

In this section, we aim to interpret the differences between the models learned with and without Editable Training. First, we investigate which inputs are most affected when the model is edited on a sample that belongs to each particular class. Based on Figure 2 (left), we conclude that edits of baseline model cause most drawdown on samples that belong to the same class as the edited input (prior to edit). However, this visualization loses information by reducing edits to their class labels.

In Figure 2 (middle) we apply t-SNE (van der Maaten & Hinton (2008)) to analyze the structure of the "edit space". Intuitively, two edited versions of the same model are considered close if they make similar predictions. We quantify this by computing KL-divergence between the model's predictions before and after edit for each of 10.000 test samples. These KL divergences effectively form a 10.000-dimensional model descriptor. We compute these descriptors for 4.500 edits applied to models trained with and without Editable Training. These vectors are then embedded in two-dimensional space with the t-SNE algorithm. We plot the obtained charts on Figure 2 (middle), with point colors denoting original class labels of edited images. As expected, the baseline edits for images of the same class are mapped to close points.

In turn, Editable Training does not always follow this pattern: the edit clusters are formed based on both original and target labels with a highly interlinked region in the middle. Combined with the fact that Editable Training has a significantly lower drawdown, this lets us hypothesize that with Editable Training neural networks learn representations where edits affect objects of the same original class to a smaller extent. Conversely, the t-SNE visualization lacks information about the true dimensionality

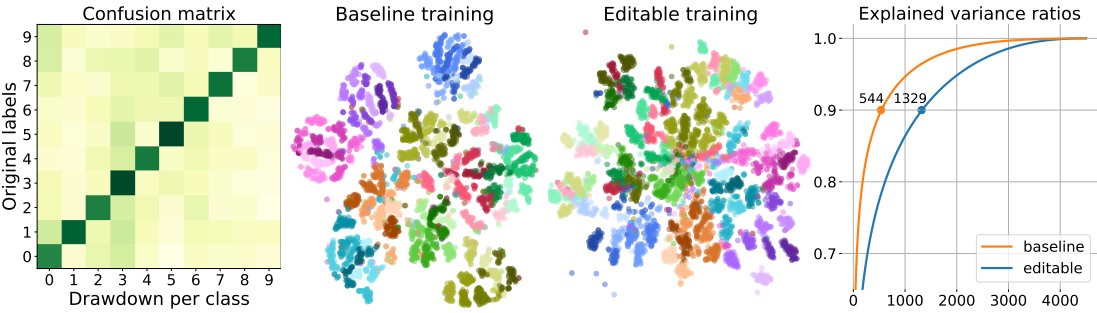

Figure 2: Edited model visualizations **(Left)** Confusion matrix of baseline model: rows correspond to editing images belonging to each of 10 classes; columns represent drawdowns per individual class. **(Middle)** t-SNE visualizations. Point color represents original class labels; brightness encodes edit targets **(Right)** The proportion of explained variance versus the number of components.

of the data manifold. To capture this property, we also perform truncated SVD decomposition of the same matrix of descriptors. Our main interest is the number of SVD components required to explain a given percentage of data variance. In Figure 2 (right) we report the explained variance ratio for models obtained with and without Editable Training. These results present evidence that Editable Training learns representations that exploit the neural network capacity to a greater extent.

### 4.3 EDITABLE FINE-TUNING FOR LARGE SCALE IMAGE CLASSIFICATION

Section 4.1 demonstrates the success of Editable Training on the small CIFAR-10 dataset. However, many practical applications require training for many weeks on huge datasets. Re-training such model for the sake of better edits may be impractical. In contrast, it would be more efficient to start from a pre-trained model and fine-tune it with Editable Training.

Here we experiment with the ILSVRC image classification task (Deng et al. (2009)) and consider two pre-trained architectures: smaller ResNet-18 and deeper DenseNet-169 (Huang et al. (2016)) networks. For each architecture, we start with pre-trained model weights[2] and fine-tune them on the same dataset with Editable Training. More specifically, we choose the training objective $\mathcal{L}_{base}(\theta)$ as KL-divergence between the predictions of the original network and its fine-tuned counterpart. Intuitively, this objective encourages the network to preserve its original classification behavior, while being trained to allow local edits. Similar to Section 4.1, the editor functions are only allowed to modify a subset of neural network layers. We experiment with two choices of such subsets. First, we try to edit a pre-existing layer in the network. Namely, we select the third out of four "chains" in both architectures. In the second experiment, we augment each architecture with an extra trainable layer after the last convolutional layer. We set an extra layer to be a residual block with a 4096-unit dense layer, followed by ELU activation (Clevert et al. (2015)) and another 1024-unit dense layer.

The evaluation is performed on $N=1000$ edits with random target class. We measure the drawdown on the full ILSVRC validation set of 50.000 images. We use the SGD optimizer with momentum $\mu=0.9$. We set the learning rate to $10^{-5}$ for the pre-existing layers and $10^{-3}$ for the extra block. The ImageNet training data is augmented with random resized crops and random horizontal flips.

Our baselines for this task are the pre-trained architectures without Editable Fine-Tuning. However, during experiments, we noticed that minimizing the KL-divergence $\mathcal{L}(\theta)$ has a side-effect of improving validation error. We attribute this improvement to the self-distillation phenomenon (Hinton et al. (2015); Furlanello et al. (2018)). To disentangle these two effects, we consider an additional baseline where the model is trained to minimize the KL-divergence without Editable Training terms. For fair comparison, we also include baselines that edit an extra layer. This layer is initialized at random for the pre-trained models and fine-tuned for the models trained with distillation.

---

[2]We use publicly available pre-trained models from `https://github.com/pytorch/vision`.

| Model Architecture | Training Procedure | Editable Layers | Test Error Rate | Mean Drawdown | Success Rate | Num Steps |
|---|---|---|---|---|---|---|
| ResNet18 | Pre-trained | Chain 3 | 30.95% | 3.89% | 99.8% | 3.582 |
| | Pre-trained | Extra layer | 30.95% | 9.18% | 100% | 4.272 |
| | Distillation | Extra layer | 30.75% | 2.80% | 100% | 2.63 |
| | Editable | Chain 3 | 30.53% | 3.78% | 99.8% | 3.616 |
| | Editable | Extra layer | 30.61% | **0.57%** | 100% | 3.388 |
| DenseNet169 | Pre-trained | Chain 3 | 25.49% | 5.20% | 100% | 2.551 |
| | Pre-trained | Extra layer | 25.47% | 9.05% | 100% | 3.874 |
| | Distillation | Extra layer | 24.33% | 1.67% | 100% | 2.822 |
| | Editable | Chain 3 | 24.32% | 4.47% | 100% | 2.556 |
| | Editable | Extra layer | 24.38% | **0.96%** | 100% | 2.970 |

Table 4: Editable Training on the ImageNet dataset with RMSProp editor function.

The results in Table 4 show that Editable Training can be effectively applied in the fine-tuning scenario, achieving the best results with an extra trainable layer. In all cases Editable Fine-Tuning took under $48$ hours on a single GeForce 1080 Ti GPU while a single edit requires less than $150$ ms.

### 4.3.1 REALISTIC EDIT TASKS WITH NATURAL ADVERSARIAL EXAMPLES

In all previous experiments, we considered edits with randomly chosen target class. However, in many practical scenarios, most of these edits will never occur. For instance, it is far more likely that an image previously classified as "plane" would require editing into "bird" than into "truck" or "ship". To address this consideration, we employ the Natural Adversarial Examples (NAE) data set by Hendrycks et al. (2019). This data set contains $7.500$ natural images that are particularly hard to classify with neural networks. Without edits, a pre-trained model can correctly predict less than $1\%$ of NAEs, but the correct answer is likely to be within top-100 classes ordered by predicted probabilities (see Figure 5 left).

The next set of experiments quantifies Editable Training in this more realistic setting. All models are evaluated on a sample of $1.000$ edits, each corresponding to one Natural Adversarial Example and its reference class. We measure the drawdown from each edit on $50.000$ ILSVRC test images. We evaluate best techniques from Section 4.3 and their modifications that account for NAEs:

- **Editable Training: Random** — model trained to edit on random targets from the uniform distribution, same as in Table 4. Compared to the same pre-trained and distilled baselines.

- **Editable Training: Match Ranks** — model trained to edit ImageNet training images with targets sampled based on their rank under NAE rank distribution (see 5, left).

- **Editable Training: Train on NAE** — model trained to edit $6.500$ natural adversarial examples. These NAEs do not overlap with $1.000$ NAE examples used for evaluation.

The results in Table 5 (top-left) show that Editable Training significantly reduces drawdown for NAEs even when trained with random targets. However, accounting for the distribution of target classes improves locality even further. Surprisingly enough, training on $6.500$ actual NAEs fares no better than simply matching the distribution of target ranks.

For the final set of evaluations, we consider two realistic scenarios that are not covered by our previous experiments. First, we evaluate whether edits performed by our method generalize to substantially similar inputs. This behavior is highly desirable since we want to avoid the edited model repeating old mistakes in a slightly changed context. For each of $1,000$ NAEs, we find the most similar image from test set based on InceptionV3 embeddings (Szegedy et al., 2015). For each such pair, we edit 5 augmentations of the first image and measure how often the model predicts the edited class on 5 augmentations of the second image. A model trained with random edits has an accuracy of 86.7% while "Editable + Match Ranks" scores 85.9% accuracy. Finally, we evaluate if

| Training Procedure | Test Error | Drawdown | Success Rate | Num Steps |
|---|---|---|---|---|
| Baseline Training | | | | |
| Pre-trained | 30.99% | 4.54% | 100% | 3.822 |
| Distillation | 30.75% | 1.62% | 100% | 2.192 |
| Editable Training | | | | |
| Random edits | 30.79% | 0.314% | 100% | 2.594 |
| Match ranks | 30.76% | **0.146%** | 100% | 2.149 |
| Train on NAE | 30.86% | **0.167%** | 100% | 2.236 |

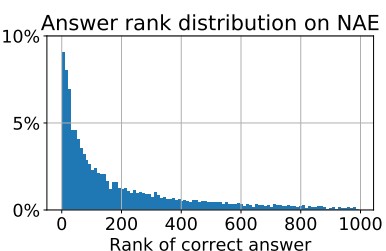

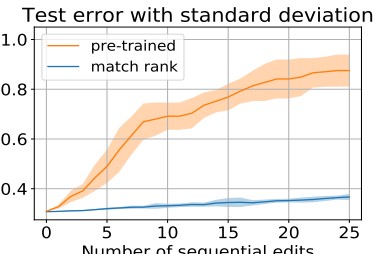

Table 5: Editing Natural Adversarial Examples for ResNet18: **(Top-Left)** Editor effectiveness when editing $N = 1000$ NAEs; **(Top-Right)** Reference class rank distribution for baseline model, **(Bottom-Right)** Error rate for edit sequences, ResNet18 baseline and Match Ranks. Pale areas indicate std. deviation over 10 runs.

our technique can perform multiple edits in a sequence. Figure 5 (bottom-left) demonstrates that our approach can cope with sequential edits without ever being trained that way.

### 4.4 EDITABLE TRAINING FOR MACHINE TRANSLATION

The previous experiments focused on multi-class classification problems. However, Editable Training can be applied to any task where the model is trained by minimizing a differentiable objective. Our final set of experiments demonstrates the applicability of Editable Training for machine translation. We consider the IWSLT 2014 German-English translation task with the standard training/test splits (Cettolo et al. (2015)). The data is preprocessed with Moses Tokenizer (Koehn et al. (2007)) and converted to lowercase. We further apply the Byte-Pair Encoding with 10.000 BPE rules learned jointly from German and English training data. Finally, we train the Transformer (Vaswani et al. (2017)) model similar to transformer-base configuration, optimized for IWSLT De-En task[3].

Typical machine translation models use beam search to find the most likely translation. Hence we consider an edit to be successful if and only if the log-probability of target translation is greater than log-probability of any alternative translation. So, $l_e(\hat{\theta}) = \max_{y_i} \log p(y_i|s, \hat{\theta}) - \log p(y_0|s, \hat{\theta})$, where $s$ is a source sentence, $y_0$ denotes target translation and $\{y_i\}_{i=1}^k$ are alternative translations. During training, we approximate this by finding $k{=}32$ most likely translations with beam search using the Transformer model trained normally on the same data. The edit targets are sampled from the same model by sampling with temperature $\tau{=}1.2$. The resulting edit consists of three parts: a source sentence, a target translation and a set of alternative translations.

We define $\mathcal{L}_{loc}$ as KL-divergence between the predictions of the original and edited model averaged over target tokens, $\mathcal{L}_{loc} = \underset{x,y \in D}{E} \frac{1}{|y|} \sum_t D_{KL}(p(y_t|x, y_{0:t}, \theta) \,||\, p(y_t|x, y_{0:t} Edit_{\alpha}^k(\theta, l_e)))$, where $D$ is a data batch, $x$ and $y$ are the source and translation phrases respectively, $y_{0:t}$ denotes a translation prefix. The $Edit$ function optimizes the final decoder layer using RMSProp with hyperparameters tuned as in Section 4.1. The results in Table 6 show that Editable Training produces a model that matches the baseline translation quality but has less than half of its drawdown.

| Training Procedure | Test BLEU | BLEU Drawdown | Success rate | Num Steps |
|---|---|---|---|---|
| Baseline training, $\alpha{=}10^{-3}$ | 34.77 | 0.76 | 100% | 2.35 |
| Editable, $c_{loc}{=}100, \alpha{=}10^{-3}$ | 34.80 | 0.35 | 100% | 3.07 |
| Editable, $c_{loc}{=}100, \alpha{=}3 \cdot 10^{-4}$ | 34.81 | **0.17** | 100% | 5.5 |

Table 6: Evaluation of editable Transformer models on IWSLT14 German-English translation task.

---

[3]We use Transformer configuration "transformer_iwslt_de_en" from Fairseq v0.8.0 (Ott et al. (2019))

## 5 CONCLUSION

In this paper we have addressed the efficient correction of neural network mistakes, a highly important task for deep learning practitioners. We have proposed several evaluation measures for comparison of different means of model correction. Then we have introduced Editable Training, a training procedure that produces models that allow gradient-based editing to address corrections of the model behaviour. We demonstrate the advantage of Editable Training against reasonable baselines on large-scale image classification and machine translation tasks.

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
