# OpenReview forum: "Editable Neural Networks"
_ICLR.cc/2020/Conference — Accept (Poster)_

### Official Review · AnonReviewer2 · 2019-10-23
**Official Blind Review #2**

**Rating:** 6

**Review:**


-----------
Update after the authors' response: the authors addressed some of my concerns and presented some new results that improve the paper. I am therefore upgrading my score to "Weak Accept".
-----------

This paper proposes a way to effectively "patch" and edit a pre-trained neural network's predictions on problematic data points (e.g. where mistakes on these data points can lead to serious repercussions), without necessarily re-training the network on the entire training set plus the problematic samples. More concretely, the edit operation on the problematic samples are done using a few steps of stochastic gradient descent.

Furthermore, the paper modifies the training procedure to make it easier for the model to perform effective patching, based on three criteria: (i) reliability (i.e. obeying the constraints specified in the edit), (ii) locality (i.e. only minimally changing the model parameters to account for the problematic samples, while ensuring the model's performance on unrelated samples remains consistent and does not degrade), and (iii) efficiency in terms of runtime and memory. To this end, the paper uses a loss term that incorporates these criteria, as weighted by interpolation coefficient hyper-parameters. The edit operation is differentiable with respect to the model parameters, which takes a similar form as the MAML approach and similarly necessitates computing second-order derivatives with respect to the model parameters. Experiments are done on CIFAR-10 toy experiments, large-scale image classification with adversarial examples, and machine translation.

Overall, while the idea of patching neural network predictions is really interesting, I have several concerns regarding the paper in its current form. I am therefore recommending a "Weak Reject". I have listed the pros and cons of this paper below, and look forward to the authors' response to the concerns I have raised.

Pros:
1. The paper is well-written, and clearly motivates the problem and why it is important.

2. The proposed approach is explained very clearly and is easy to understand.

3. The paper correctly identifies the use of distillation loss (Table 4) as a potential confound, and runs a distillation baseline without editable training as an additional baseline. This improves the thoroughness of the experiments, and ensures that the gains can really be attributed to the proposed editable training objective.

4. The paper makes interesting connections to other problems where the editable training approach can potentially be useful, such as catastrophic forgetting and adversarial training.

Cons:
1. I still have my doubts about the locality constraint. If the model makes a mistake on some problematic samples, what we want is not just fixing the model's predictions only on these problematic samples, but also on other samples that share a substantial degree of similarity to the problematic samples, so that the model can avoid making the same broader type of mistakes in the future. On the surface, the locality constraint seems to do the opposite, since it confines the impact of the edit operation only to the problematic samples. In contrast, the alternative of re-training the neural network on the full training set augmented with the problematic samples can potentially overcome this problem (since the entire model parameters are updated, correcting the model's mistakes on the problematic samples would help the model avoid making the same mistakes on other similar samples), although of course computationally much more expensive to do.

2. Related to Point 1 above, the equation defining the locality constraint (Eq. 4) seems to require an expectation of x that is drawn from p(x). How is this quantity computed? Does the model assume that x comes from the empirical distribution? Also, the "control set" (bottom of page 2) is not well defined. This is very important, since the control set defines what examples should (or should not) be changed by the editing operations. These two points should be clarified further.

3. Related to Points 1 and 2 above, I am not sure whether lower "Drawdown" is an indication of better editable training. As mentioned before, ideally the model should not only "patch" its mistakes on problematic samples, but also on samples that are substantially similar to these problematic samples. Naturally this might result in more differences from the initial model, which also means less locality (i.e. higher locality/lower drawdown may not necessarily be a good evaluation metric to evaluate edit effectiveness).

4. The paper contains a substantial set of experiments and analysis based on toy experiments on CIFAR-10, where the labels of some examples are randomly swapped (i.e. random noise). I think this setup is dangerous, since the paper is drawing conclusions based on how well editable training can fit random noise! Instead, all the experiments should be done on real use cases (e.g. adversarial training or catastrophic forgetting). It is unfortunate that the paper has more content for the toy experiments (2.5 pages) than for the adversarial example experiment in image recognition (1.5 pages) or machine translation experiment (0.5 page).

5. The hyper-parameter experiments for selecting the learning rates can be put in the Appendix instead, rather than taking up nearly half a page in the main text (e.g. Table 1). This would leave more space for experiments, more explanation, intuition, analysis, etc.

6. The improvements (in terms of both image classification error rates or machine translation BLEU score) afforded by editable training are quite small. For instance, in the machine translation experiment, Table 6 indicates that editable training only leads to a 0.04 BLEU improvement. This is really small by standard machine translation literature, and can very well be explained by randomness in hyper-parameter initialisation, etc., rather than a better training procedure. It is hard to draw any conclusions based on such small gains.

More minor points:
1. The use of bold in the Tables are not very consistent. For instance, in Table 3, for column "Test Error Rate", 6.31% is in bold, even though baseline training has an accuracy of 6.3%, which is lower (and thus better). Also, in the "test error drawdown" column, 0.86% is in bold, even though there is a lower entry of 0.65%.

2. Missing citation to the work of Furlanello et al. (2018) for self-distillation.

References
Tommaso Furlanello, Zachary C. Lipton, Michael Tschannen, Laurent Itti, and Anima Anandkumar. Born Again Neural Networks. In Proc. of ICML 2018.

**Experience Assessment:**

I do not know much about this area.

**Review Assessment: Checking Correctness Of Derivations And Theory:**

I assessed the sensibility of the derivations and theory.

**Review Assessment: Checking Correctness Of Experiments:**

I assessed the sensibility of the experiments.

**Review Assessment: Thoroughness In Paper Reading:**

I read the paper at least twice and used my best judgement in assessing the paper.

---

> ### Author Response · Authors · 2019-11-06
> **To Reviewer 2: We try to address your concerns.**
>
> Thank you for a thorough examination of our paper. We address the items from your Cons list below.
>
> Cons
> (1, 3). This is an important point that we have addressed in a new revision of the paper. Indeed, the locality (and hence Drawdown) does not capture for the fact that we *want* the model to change its prediction on inputs that are substantially similar to the edited example.
>
> We have conducted an experiment that evaluates the ability of Editable Training to generalize to substantially similar inputs. In this experiment, we use the same 1000 Natural Adversarial Examples as in Table 5. For each example, we edit 5 random augmentations of this image and then evaluate accuracy on another 100 random augmentations. In particular, we use randomized crop and horizontal flip augmentations similar to previous ILSVRC experiments. A model trained with random edits was able to correctly classify 95.2% and the "Editable + Match Ranks" model scored 96.0%.
>
> These experiments demonstrate that the editor function does generalize to similar images. Indeed, addressing this issue is a valuable addition to the submission.
>
> 2. The expectation in Eq. 4 is estimated with monte-carlo: we compute the average KL divergence between the two models over a minibatch of images from the training set. The Editable Training supports any problem-specific choice of the control set, the specific choices are discussed in the experimental section (4.1). In particular, we use the entire test set to measure drawdown.
>
> 4. While we agree that Sections 4.3 and 4.4 are of greater practical importance, we would like to defend our choice of CIFAR10 for preliminary experiments. Editable Training on CIFAR10 is not affected by factors like structured prediction (MT) or fine-tuning (ImageNet). This allows us to decouple the effects of Editable Training from these factors when conducting experiments. Last but not least, CIFAR10 allowed us to conduct experiments faster and make better use of limited computational resources.
>
> 6. We are afraid that there was a misunderstanding: Editable Training does *not* aim to improve Classification Error or BLEU score and we make no such claims in the submission. Instead, the proposed method improves the model’s capacity for edits without harming other metrics. The differences in Error/BLEU from Tables 4-6 are reported only to support our claim that Editable Training does not decrease the task-specific quality of the model.
>
> Minor points
> (1, 2) Thank you, we have fixed both issues in a new revision.

---

> > ### Comment · AnonReviewer2 · 2019-11-12
> > **Reply to the authors' response**
> >
> > Thank you for your detailed response, and for incorporating the comments in the latest version of the paper. I find the new version to be better, although I still have some follow-up comments.
> >
> > On comments (1, 3), it is indeed encouraging that an edit operation makes the model change its predictions for similar inputs. However, I am not entirely convinced by the experimental setup, since this is only shown for different augmentations (in my understanding this means rotate, etc.) of *the same image*. Clearly there is a lot of overlap between two augmentations of the same image, and the fact that the edit operation also changes the model's predictions for these augmentations is not really surprising. It would be better if this is shown for more natural examples (e.g. images that belong to the same class label/qualitatively look similar).
> >
> > On comment (4), I agree that experiments on CIFAR10 are good for abstracting away from the full complexities of other problems. My issue is more that the CIFAR10 experiments mostly show how well the edit operations are able to fit *noise*, since the true labels are randomly swapped, which is not a very good measure of edit usefulness in practice. For instance, using the natural adversarial examples experiment on CIFAR10 is a good step in this direction (i.e. still using CIFAR10, but with a real use case rather than swapping the random labels).

---

> > > ### Author Response · Authors · 2019-11-13
> > > **To Reviewer 2**
> > >
> > > Thank you for the prompt reply.
> > >
> > > (1, 3)
> > > We agree that it would be interesting to evaluate editor consistency for similar images. However, in many cases, a random pair of images of the same class will look significantly different. Therefore, like you proposed, we focus on on semantically similar images.
> > >
> > > For each NAE we find the nearest image from ILSVRC test set based on InceptionV3 embeddings. For each pair of semantically similar images we edit the first image and observe if the second one matches the edited class. In this evaluation, a model trained with random edits classified 86.7% of pairs correctly and the "Editable + Match Ranks" model classified 85.9%. We will replace the experiment with augmentations by this refined version.
> > >
> > > (4)
> > > We would also like to use a natural edit distribution in our CIFAR10 experiments. However, Natural Adversarial Examples is a dataset of ImageNet-sized images that was manually labeled with ImageNet classes and has almost no intersection with CIFAR10. While we could downscale the images, theres nothing we can do about the classes.
> > >
> > > However, we argue that our experiments with random labels *are* useful in practice. In Section 4.3.1 where we do have natural adversarial examples, we observe that Editable Training with random labels actually improves model's ability to edit NAEs. Please refer to Table 5 for the exact numbers.

---

> > > > ### Comment · AnonReviewer2 · 2019-11-14
> > > > **Reply**
> > > >
> > > > Thank you for the prompt response and the clarification. These results improve the paper, and I will take these into consideration for my final review.

---

### Official Review · AnonReviewer3 · 2019-10-26
**Official Blind Review #3**

**Rating:** 3

**Review:**

The authors propose Editable Training that edits/updates a trained model using a model-agnostic training technique. Editable training is able to correct mistakes of trained models without retraining the whole model nor harming the original performance. This is attained via meta-learning techniques to avoid catastrophic forgetting, and an editor function to promise mistake correction. The major contribution is a model-agnostic editable training process that is applicable to various neural nets. This paper has brought attention to mistake correction problem in neural networks and proposes a simple and concise solution. In addition, extensive experiments on both small and large-scale image classification and machine translation tasks demonstrate the effectiveness of different editor functions. Overall, this paper is well-written with extensive experimental results. Below are a few concerns I have to the current status of the paper.

1.	It would be interesting to discuss if how a good editor function changes over different models, problems, or even l_e’s. In addition.
2.	In general, a DNN needs to be “edited”/”fixed”, when the training data used are not sufficient, and /or the incoming testing data have a different distribution from the training data. In the latter case, say, if the distribution of new data is significantly different from the training data used so far, it may be worth of re-train the model rather than attempting to “fix” the network. There should be a trade-off between “fixable” vs “not-fixable”. It is unclear how this trade-off is modeled/discussed in the paper.


**Experience Assessment:**

I have published one or two papers in this area.

**Review Assessment: Checking Correctness Of Derivations And Theory:**

I carefully checked the derivations and theory.

**Review Assessment: Checking Correctness Of Experiments:**

I carefully checked the experiments.

**Review Assessment: Thoroughness In Paper Reading:**

I read the paper thoroughly.

---

> ### Author Response · Authors · 2019-11-06
> **To Reviewer 3: We try to address your concerns.**
>
> Thank you for time and your comments. Below, we try to address your concerns.
>
> [if/how a good editor function changes over different models, problems, or even l_e’s]
>
> We agree that this is an interesting direction to investigate. As an additional experiment, we have evaluated all editor functions for a different architecture (namely, DenseNet40 with growth=12) on the CIFAR10 dataset. The test error rate of this model is 5.3%. The results are reported in the table below. As you can see, the RMSProp editor function is optimal for this architecture as well.
>
> | editor func. | Gradient Descent | Scaled GD | RProp  | RMSProp  | Momentum | Adam   |
> |-----------------|--------------------------|---------------|-----------|---------------|------------------|-----------|
> | drawdown  |      3.1%                   |   2.31%       | 1.84%  |   1.65%      |  2.70%           |  4.15%  |
> | success rate|     99.8%                 |  100%         | 100%   |   100%       |  100%            |  98.1%   |
> | avg # steps |     3.48                     |  3.68           |  2.57    |  3.06          |  3.86              |  4.08      |
>
>
> For this table we used the same parameter tuning as for Table 1 in the submission. Unfortunately, such exhaustive evaluation is not feasible for large-scale tasks. However, we have observed that the RMSProp editor consistently outperforms Gradient Descent on both ImageNet and Machine Translation tasks.
>
> Our results agree with an intuition that Resilient Backpropagation-based methods (e.g. RProp and RMSProp) are less likely to make drastic changes in the model behavior[1]. For similar reason such methods are widely used e.g. in  Adversarial Attacks[2], where large changes of images are not desirable.
>
> [ it may be worth to re-train the model rather than attempting to “fix” the network ]
> Editable Training is motivated by a usage scenario where one needs to quickly adjust model behavior - fix one or a few sensitive mistakes without changing predictions globally.
> In contrast, if one needs to change model behavior on a large subset of inputs (e.g. add an entirely new class or domain) it is indeed better to re-train the model from scratch with the extra training samples.
>
> The need for Editable Training is even more notable if the model is deployed on mobile, IoT or embedded systems. Imagine an image classification system embedded into a camera with limited internet access (e.g. a satellite). When editing such a system, you can generally expect its hardware to take a single image and run a few gradient descent steps in a reasonable amount of time. In turn, re-training the system from scratch would require a large dataset and a GPU server.
>
> [1] Martin Riedmiller und Heinrich Braun: Rprop - A Fast Adaptive Learning Algorithm. Proceedings of the International Symposium on Computer and Information Science VII, 1992
>
> [2] Goodfellow, Ian J., Jonathon Shlens and Christian Szegedy. “Explaining and Harnessing Adversarial Examples.” CoRR abs/1412.6572 (2014)

---

> > ### Comment · AnonReviewer3 · 2019-11-15
> > **Thanks for the response.**
> >
> > Thank you for your detailed responses. I will consider these in the final review.

---

### Official Review · AnonReviewer1 · 2019-10-27
**Official Blind Review #1**

**Rating:** 8

**Review:**

This paper makes a convincing case for improving the usefulness of production level, large scale models by making them quickly editable without extensive retraining. A standard meta-learning method called MAML is used to augment the initial training phase of such models, or to "patch" them later on. The paper demonstrates effectiveness for both image classification and machine translation tasks, covering a wide range of relevant scenarios.

I recommend acceptance because:
- The paper considers a real issue for production models which are becoming widespread, and retraining for targeted modifications is impractical.
- Experiments are consistently well designed and executed. The proposed benchmarks are relevant for future work as well.
- The paper is clear and easy to read.

The method proposed in [1] is very similar, even if it is used in the continual learning settings. I would have liked to see some continual learning solutions used as baselines and/or combined with the proposed method, as they do manage to improve performance in [1]. Some candidates would be L2-regularization and EWC [2].

References:
[1] Khurram Javed, Martha White. Meta-Learning Representations for Continual Learning. CORR 2019. https://arxiv.org/pdf/1905.12588.pdf
[2] James Kirkpatrick et al. Overcoming catastrophic forgetting in neural networks. PNAS 2017. https://arxiv.org/pdf/1612.00796.pdf

**Experience Assessment:**

I have published one or two papers in this area.

**Review Assessment: Checking Correctness Of Derivations And Theory:**

N/A

**Review Assessment: Checking Correctness Of Experiments:**

I assessed the sensibility of the experiments.

**Review Assessment: Thoroughness In Paper Reading:**

I read the paper at least twice and used my best judgement in assessing the paper.

---

> ### Author Response · Authors · 2019-11-07
> **To Reviewer 1: We address your concerns.**
>
> Thank you for your valuable comments. Below we report the experimental evaluation of EWC and L2 editor functions on the CIFAR10 dataset. Namely, we adapt the implementation [1] to perform edits as the second “task” after training normally. We tune $lambda \in [0, 10]$ similarly to other hyperparameters (optimizer, editable layers, etc.).
>
> |  Training | Editor Function |  Layers  |  Error | Drawdown | Success Rate | # Steps |
> |--------------|----------------------|-------------|---------|-----------------|-------------------|------------|
> |  Baseline |       GD + L2       | Chain 3  |  6.3%  |    2.54%       |     99.3%         |   3.94     |
> |  Baseline | RMSProp + L2  | Chain 3   |  6.3%  |   1.39%        |     100%         |    3.87    |
> |  Baseline |      GD+EWC      | Chain 3   |  6.3%  |   1.92%        |     100%         |    3.88   |
> |  Baseline | RMSProp+EWC | Chain 3   |  6.3%  |   1.24%        |     98.1%        |    4.03   |
> |======================Previous experiments for comparison=================|
> |  Baseline |      RMSProp     | Chain 3   |  6.3%  |   1.77%        |     100%        |    3.11   |
> |  Editable  |      RMSProp     | Chain 3   |  6.31% |   0.86%        |     100%        |    4.13   |
>
>
> The experiments demonstrate that Elastic Weight Consolidation improves editor Drawdown for both GD and RMSProp. While EWC is still inferior to Editable Training (0.86% vs 1.24%), we believe that in future these two methods can be combined to improve editor functions even further. Meanwhile, we have added EWC into Section 4.1 in a new revision.
>
> [1] https://github.com/moskomule/ewc.pytorch

---

### Author Response · Authors · 2019-11-08
**Common answer to the reviewers.**

We thank the reviewers for their comments. We have uploaded a new version and we summarize the changes below. We also address the individual concerns of each reviewer in separate comments.

1. Section 4.1: Added comparison to Elastic Weight Consolidation;
2. Section 4.3.1: Added an experiment that evaluates how edited models generalize to substantially similar inputs;
3. Changed boldness in tables. Now the results in bold always indicate experiments with lowest drawdown;
4. Cited Furlanello et al. (2018) for self-distillation.
5. Slightly changed paragraph composition to improve visual presentation.

We are ready to address new concerns till the end of Discussion Period.

---

### Decision · Program_Chairs · 2019-12-19

**Decision:**

Accept (Poster)

**Comment:**

This paper proposes a method which patches/edits a pre-trained neural network's predictions on problematic data points. They do this without the need for retraining the network on the entire data, by only using a few steps of stochastic gradient descent, and thereby avoiding influencing model behaviour on other samples. The post patching training can encourage reliability, locality and efficiency by using a loss function which incorporates these three criteria weighted by hyperparameters. Experiments are done on CIFAR-10 toy experiments, large-scale image classification with adversarial examples, and machine translation. The reviews are generally positive, with significant author response, a new improved version of the paper, and further discussion. This is a well written paper with convincing results, and it addresses a serious problem for production models, I therefore recommend that it is accepted.